# Tomato Twisted Leaf Virus: A Novel Indigenous New World Monopartite Begomovirus Infecting Tomato in Venezuela

**DOI:** 10.3390/v11040327

**Published:** 2019-04-04

**Authors:** Gustavo Romay, Francis Geraud-Pouey, Dorys T. Chirinos, Mathieu Mahillon, Annika Gillis, Jacques Mahillon, Claude Bragard

**Affiliations:** 1UCLouvain, Earth and Life Institute, Applied Microbiology-Phytopathology, Croix du Sud 2-L07.05.03, 1348 Louvain-la-Neuve, Belgium; mathieu.mahillon@uclouvain.be; 2La Universidad del Zulia (LUZ), Unidad Técnica Fitosanitaria, Maracaibo 4005, Estado Zulia, Venezuela; fgeraudp@gmail.com; 3Facultad de Ingeniería Agronómica, Universidad Técnica de Manabí, Manabí 130105, Ecuador; dtchirinos@gmail.com; 4UCLouvain, Earth and Life Institute, Applied Microbiology-Laboratory of Food and Environmental Microbiology, Croix du Sud 2-L7.05.12, 1348 Louvain-la-Neuve, Belgium; annika.gillis@uclouvain.be (A.G.); jacques.mahillon@uclouvain.be (J.M.)

**Keywords:** begomovirus evolution, *Geminiviridae*, *Solanum lycopersicum*, tomato crop

## Abstract

Begomoviruses are one of the major groups of plant viruses with an important economic impact on crop production in tropical and subtropical regions. The global spread of its polyphagous vector, the whitefly *Bemisia tabaci*, has contributed to the emergence and diversification of species within this genus. In this study, we found a putative novel begomovirus infecting tomato plants in Venezuela without a cognate DNA-B component. This begomovirus was genetically characterized and compared with related species. Furthermore, its infectivity was demonstrated by agroinoculation of infectious clones in tomato (*Solanum lycopersicum*) and *Nicotiana benthamiana* plants. The name Tomato twisted leaf virus (ToTLV) is proposed. ToTLV showed the typical genome organization of the DNA-A component of New World bipartite begomoviruses. However, the single DNA component of ToTLV was able to develop systemic infection in tomato and *N. benthamiana* plants, suggesting a monopartite nature of its genome. Interestingly, an additional open reading frame ORF was observed in ToTLV encompassing the intergenic region and the coat protein gene, which is not present in other closely related begomoviruses. A putative transcript from this region was amplified by strand-specific reverse transcription-PCR. Along with recent studies, our results showed that the diversity of monopartite begomoviruses from the New World is greater than previously thought.

## 1. Introduction

Over the course of the last few decades, members of the genus *Begomovirus* have remained serious threats to tomato production in tropical and subtropical regions across the world. Begomoviruses represent the largest group of plant viruses and belong to the family *Geminiviridae* along with members of eight different genera: *Becurtovirus*, *Capulavirus*, *Curtovirus*, *Eragrovirus*, *Grablovirus*, *Mastrevirus*, *Topocuvirus*, and *Turncurtovirus* [1]. One of the most important factors mediating the ever-increasing emergence and dissemination of begomoviruses is the global spread of its vector, the whitefly *Bemisia tabaci* [2]. The genome of begomoviruses has one or two circular single-stranded DNA components, referred to as DNA-A and DNA-B [3]. The DNA-A of bipartite begomoviruses is homologous to the genome of monopartite begomoviruses and encodes the capsid protein (open reading frame (ORF) AV1/V1), the AV2/V2 protein (ORF AV2/V2), the replication-associated protein (ORF AC1/C1), the transcriptional activator protein (ORF AC2/C2), the replication enhancer protein (ORF AC3/C3), and the AC4/C4 protein (ORF AC4/C4), while the DNA-B encodes the nuclear shuttle protein (ORF BV1) and the movement protein (ORF BC1) [1]. For taxonomical purposes, the full-length sequence of DNA-A, or its homologous DNA of monopartite begomoviruses, is used to classify and distinguish species into their begomovirus group considering 91% of nucleotide identity as the demarcation threshold [3].

Begomoviruses can be separated in two geographically distant groups, the New World (NW) group and the Old World (OW) group. In general, NW begomoviruses have a bipartite genome, while the majority of OW begomoviruses (ca. 85%) display a monopartite genome [2]. Both DNA-A and DNA-B components of bipartite begomoviruses are required for systemic infection [3]: while DNA-A is responsible for replication and transcription functions, DNA-B is implicated in the viral movement [4]. Unlike bipartite begomoviruses, only the single DNA component of monopartite begomoviruses is essential to develop systemic infection [4]. Another difference between both geographic groups is the lack of the AV2 gene in the genome of NW begomoviruses [1].

Although NW begomoviruses had been thought to have only bipartite genomes, a new tomato begomovirus containing a single DNA component was reported in Peru in 2011 [5]. This virus was named *Tomato leaf deformation virus* (ToLDeV) and its infectivity was demonstrated through development of several infectious clones [6,7]. In this study, a putative new begomovirus species infecting tomato plants was found in Venezuela containing a single DNA component. Further assays to fulfill Koch’s postulates were conducted with this novel virus and demonstrated the monopartite nature of its genome.

## 2. Materials and Methods

### 2.1. Sample Collection and Begomovirus Detection

In July 2005, during a survey of begomoviruses carried out in Venezuela [8], leaf samples were collected in Zulia state from two tomato plants (samples 2005-07-22-01 and 2005-07-22-02; hereafter referred to as 331 and 332, respectively) exhibiting leaf deformation and stunting symptoms. Each sample was dried on a sealed receptacle containing Silicagel^®^. The dried samples were wrapped on tissue paper and stored in 2 mL tubes at −20 °C until use. Total DNA was extracted using a standard protocol [9] and PCR detection for begomoviruses was performed using the degenerate primer pairs PAL1v1978/PAR1c946 and PBLv2040/PCRc1 [10] and Taq DNA Polymerase master mix (New England BioLabs, USA). PCR conditions were as follows: 2 min at 94 °C, 35 cycles at 94, 55, and 68 °C for 30, 30, and 60 s, respectively, and a final extension of 5 min at 68 °C. PCR products were cloned in *Escherichia coli* competent cells using the pGem^®^-T Easy vector system (Promega, USA). At least five clones were sequenced for further BLAST analysis.

### 2.2. Cloning of Full-Length Viral Genomes

Using tomato samples 331 and 332, a 1.1 kb fragment was cloned and subsequently sequenced. Since they shared 99.8% of identity, only sample 332 was considered for subsequent studies. DNA from this sample was used as a template to obtain the viral sequences by rolling circle amplification (RCA) using Φ29 DNA polymerase (TempliPhi, GE Healthcare, Munich, Germany). Restriction enzymes *Bam*HI, *Eco*RI, *Hin*dIII, and *Pst*I were used to follow a protocol of restriction fragment length polymorphism (RFLP) on RCA products (RFLP-RCA) according to Haible et al. [11]. The bands were excised from agarose gel and inserted into the appropriate restriction sites in pBluescript II SK (+) (La Jolla, Stratagene, CA, USA) for subsequent transformation of *E. coli* competent cells. The cloned products were sequenced at Macrogen Inc. (Amsterdam, Netherlands). A preliminary identification of viral sequences was based on a BLAST search and three different begomovirus DNA components were observed, suggesting a mixed infection. Subsequently, the three different full-length viral clones, referred to as isolates Be1.1B, Be6.6H, and B6.7H, were completely sequenced by primer walking and assembled using BioEdit v.7.1.9 [12]. The complete nucleotide sequence of clones Be6.6H, Be6.7H, and Be1.1B were deposited in GenBank under the accession numbers MK440292, MK440293, and MK440294, respectively.

### 2.3. Pairwise Identity, Phylogenetic, and Recombination Analyses

Since the isolate Be6.6H did not show more than 90% of nucleotide identity with any begomovirus in GenBank database, it was considered for further genetic and biological studies. Isolate Be6.6H was aligned with sequences of related NW begomoviruses using MUSCLE [13] and pairwise sequence identities were estimated using the Sequence Demarcation Tool (SDT) v.1.2 [14]. The aligned sequences were analyzed to determine the optimal nucleotide substitution model and generate a maximum-likelihood phylogenetic tree (500 bootstrap replicates) using MEGA X v10.0.5 [15]. The phylogenetic tree was rooted with the OW begomovirus species *African cassava mosaic virus* (ACMV). In order to identify potential recombination events in isolate Be6.6H, putative parental begomoviruses were downloaded from GenBank based on sequence analysis with the software SWeBLAST [16]. These viruses were aligned to perform recombination analyses using the methods BoostScan, Chimera, GENCOV, MaxChi, RDP, SiScan, and 3Seq implemented in RDP4 software v.4.31 [17]. Only recombination events detected by at least five out of seven methods were considered.

### 2.4. Construction of an Infectious Clone of Isolate Be6.6H for Agroinoculation

A cognate DNA-B component of the isolate Be6.6H was found neither by PCR with begomovirus universal primers [10] nor by cloning of digested RCA products. In order to evaluate the infectivity of Be6.6H isolate, a multimeric clone was developed, as follows: the viral clone inserted in the *Hin*dIII site of pBluescript II SK (+), here referred to as pBe6.6H, was released with double digestion using enzymes *Bam*HI and *Hin*dIII. This viral fragment was inserted into pCAMBIA1300 to generate a 0.9-mer clone containing the viral intergenic region. Then, the plasmid pBe6.6H was digested with *Hin*dIII to release the full-length viral DNA and then inserted into the plasmid pCAMBIA1300 containing the 0.9-mer clone of Be6.6H isolate. The generated 1.9-mer clone was named pBe6.6Hdi. The correct orientation of the *Hin*dIII-*Hin*dIII insert was confirmed by *Sac*I restriction assays. *Agrobacterium tumefaciens* cells (strain C58C1) were transformed with plasmid pBe6.6Hdi by electroporation. PCR assays using specific primers, Be-2147F (5′-ACGGCATTGGCGTCTTTGG-3′) and Be-510R (5′-TCGTCCATCCATATCTTGCC-3′), were performed to confirm the presence of viral DNA in the bacteria.

At least six single colonies of *A. tumefaciens* harboring pBe6.6Hdi were grown at 28 °C for 48 h in 5 mL of Luria–Bertani (LB) medium supplemented with kanamycin (50 mg/L) and rifampicin (25 mg/L). Then, 50 ml of LB containing antibiotics were inoculated with 5 ml of *A. tumefaciens* culture and grown at 28 °C for 16 h with shaking. The bacteria were harvested by centrifugation at 2000 *g* for 10 min. The pellets were resuspended in 30 mL of MMA solution (10 mM MgCl_2_, 10 mM MES (pH 5.6), 100 μM acetosyringone). Then, the pellets were centrifuged at 5000 *g* for 10 min and resuspended again in 30 mL of MMA solution. The bacterial suspensions were kept at room temperature with shaking for 4 h. The final optical density at 600 nm (OD_600_) was adjusted to 0.8. Tomato cv. Moneymaker and *Nicotiana benthamiana* plantlets were infiltrated on the lower surface of young fully expanded leaves using a 1 mL needleless syringe. Ten plants of tomato and *N. benthamiana* were inoculated with the viral clone. As negative control, 10 plants of both species were infiltrated with a mock (*A. tumefaciens* carrying the empty pCAMBIA1300 vector). Three replicates of this test were done. The viral infection was checked by PCR with primers Be-2147F and Be-510R at three weeks post inoculations (wpi).

### 2.5. Genetic and Functional Analyses of a Sixth ORF in Be6.6H Isolate

A sixth viral ORF was found in both isolates from tomato samples, starting at position 84 from the cleavage site “AC” of the invariant nonanucleotide TAATATT|AC at the origin of replication. The deduced amino acid (aa) sequence was subjected to a BLASTp search. In order to determine whether a transcript is produced from this ORF, total RNA was extracted from young leaf tissue of tomato plants positive to Be6.6H isolate using TRIzol reagent (Invitrogen, USA). RT-PCR assays were performed with primers Be-089F (5′-GAGAGCGTCTGTGGAGCCT-3′) and Be-363R (5′-TGAACCTTACATGGGCCTTC-3′) and Be-2147F/Be-363R using a OneTaq^®^ One-Step RT-PCR Kit (New England BioLabs Inc., USA). Prior to RT-PCR assays, RNA extractions were treated with RQ1 RNase-Free DNase (Promega, USA) to prevent DNA viral contamination. RNA extractions, RT-PCR, and DNase treatments were performed following protocols suggested by the manufacturers.

## 3. Results

### 3.1. Sample Collection and Begomovirus Detection

Begomovirus PCR assays confirmed the viral infection in tomato plants showing leaf deformation and stunting symptoms in Zulia estate, Venezuela. The PCR tests yielded expected fragment sizes (ca. 1.1 kb). The samples 331 and 332 showed a viral sequence sharing 99.8% nucleotide identity (NI) with each other, and 85% NI with the closest related begomovirus species *Abutilon golden mosaic Yucatan virus* (AbGMYV, GenBank No. KC430935). In addition, sample 332 contained another viral sequence with the highest identity (98%) with *Tomato chlorotic leaf distortion virus* (ToCLDV, GenBank No. JN241632), indicating a mixed infection in this sample. When using degenerate primers for DNA-B, only sample 332 was positive showing the expected fragment size (ca. 500 bp). This fragment was then cloned and at least 20 clones were sequenced. They all shared 95% NI with DNA-B of ToCLDV (GenBank No. JN241633).

### 3.2. Cloning of Full-Length Viral Genome

The RFLP-RCA assays confirmed the mixed infection caused by begomoviruses in sample 332. After cloning of restricted fragments, three begomovirus DNA components were identified. The clone Be6.6H (2597 bp) exhibited the highest sequence identity (82%) with the DNA-A component of Rhynchosia golden mosaic Yucatan virus (GenBank No. EU021216), which is currently considered to be a strain of the species *Cabbage leaf curl virus* (CabLCV) [1]. The isolate Be6.6H showed the five typical ORFs found in the genome organization of DNA-A from NW bipartite begomoviruses (Figure 1A). Interestingly, an additional ORF of 342 nucleotides, named ORF 6, was noticed encompassing the intergenic region (IR) and the V1 gene that codes for the coat protein (CP) (Figure 1B). In OW begomovirus genome, the V2/AV2 gene is present in the same region as the ORF 6 (Figure 1C). However, no homology was found between the V2 or AV2 protein and the deduced aa sequence from ORF 6 of Be6.6H.

The clone Be6.7H (2623 bp) shared 99% NI with a ToCLDV DNA-A component (GenBank No. JN241632) isolated from *Capsicum chinense*, while the clone Be1.1B (2608 bp) shared the highest sequence identity (94%) with a ToCLDV DNA-B component (GenBank No. HQ201953) infecting tomato [18]. As expected, clones Be6.7H and Be1.1B shared a 172 nt common region with 93.5% of identity, indicating that they are a cognate pair of the same ToCLDV isolate. Meanwhile, Be6.6H showed a low homology (ca. 77 % NI) of this genome region with clones Be6.7H and Be1.1B.

Clone Be6.6H displayed two copies of iteron CTGGAGTC and two inverted copies (GACTCCAG) of this iteron upstream of the TATA box, while clones Be6.7H and Be1.1B only showed two copies of the same iteron sequence (TGTATTGG) (Figure 2). The iteron sequences are required for viral replication and they are specific to each begomovirus species.

### 3.3. Pairwise, Phylogenetic, and Recombination Analyses

The full-length sequence of isolate Be6.6H was compared with begomovirus sequences available in the GenBank dataset using Blastn. Table 1 shows the six most related begomoviruses to isolate Be6.6H. According to SDT pairwise scores, this isolate showed the greatest similarities, 82.4 and 82.1%, to an isolate of *Euphorbia mosaic virus* (EuMV) from Cuba (GenBank No. KU165788) and a CabLCV isolate (GenBank No. EU021216) from Mexico, respectively. Begomoviruses sharing DNA-A NI < 91% are recognized as different species [1]. Therefore, isolate Be6.6H represents a putative novel begomovirus species and the name Tomato twisted leaf virus (ToTLV) is proposed for this virus.

The phylogenetic analysis indicated that ToTLV belongs to the *Squash leaf curl virus* (SCLV) clade (Figure 3), a genetic group of begomoviruses widely distributed in the Americas [19], although forming a relative independent branch within this clade.

Recombination analysis predicted the occurrence of a recombination event in the genome of ToTLV by seven methods implemented in RDP4 software. SiScan showed the lowest *p* value (*p* = 7.195 × 10^−25^). Recombination breakpoints were detected at nucleotide positions 1942 (ORF C1) and 24 (intergenic region) of ToTLV, suggesting that this virus has a recombinant nature. *Tomato yellow mottle virus* (ToYMoV, GenBank No. KY064015) and *Jacquemontia yellow vein virus* (JacYVV, GenBank No. KY617094) were identified as putative major and minor parents, respectively. ToYMoV has been reported infecting tomato in Costa Rica [20], while JacYVV was described in Venezuela infecting *Jacquemontia tamnifolia*, a wild plant belonging to the family *Convolvulaceae* [21].

### 3.4. Infectivity of Clone Be6.6H in Tomato and N. Benthamiana Plants

*A. tumefaciens* cells, containing the 1.9-mer clone of ToTLV, were inoculated in tomato and *N. benthamiana* plants. After three wpi, the virus was detected by PCR in young leaves of both plants, indicating the systemic infection of ToTLV isolate Be6.6H. At three wpi, the transmission rate was 70% (7, 6, and 8 out of 10 plants used for each replicate, respectively). After four wpi, ToTLV-infected tomato plants showed a reduced growth, as compared with mock-inoculated plants (Figure 4A). The tomato plants also displayed twisted and curled leaves (Figure 4B). Furthermore, severe symptoms consisting of leaf deformation and a drastic reduction in the size of the new emergent leaves were observed at eight wpi (Figure 4C). In *N. benthamiana* plants, mosaic symptoms associated with ToTLV were also observed after four wpi (Appendix A) and the transmission rate was 100%.

### 3.5. Genetic Analysis of a Putative Novel Sixth ORF in Be6.6H Isolate

The genome analysis revealed a sixth viral ORF in ToTLV encompassing the intergenic and the CP coding region. BLASTp analysis did not show significant similarity to any protein in public databases. RT-PCR assays from ToTLV-infected tomato plants yielded an amplicon of ca. 300 nt when using primers Be-089F and Be-363R (Figure 5A), indicating that this ORF is potentially transcribed. The amplified fragment covered an internal region from position 2 to 100 of the putative protein coded by ORF 6. In order to exclude false positive results due to viral DNA contamination in RNA extractions, RT-PCR assays were performed using primer Be-2147F targeting the AC1 gene and primer Be-363R targeting ORF 6 (Figure 5B). Both primers were also used for PCR using DNA extractions from the same leaf tissues. RT-PCR assays did not yield any amplification products, while the PCR tests yielded the expected 850 bp fragment using the same primer pairs (Figure 5A). Although RT-PCR assays suggest that the ORF 6 is transcribed, 5′ and 3′ Rapid amplification of cDNA ends (RACE) assays and mutational analysis are required to confirm these results.

## 4. Discussion

In this study, genetic and biological analyses showed that the isolate Be6.6H is a new putative species of begomovirus according to the demarcation criteria established by the International Committee on Taxonomy of Viruses [1,3] and the name Tomato twisted leaf virus (ToTLV) is proposed based on the symptoms observed in tomato plants. ToTLV was found in single and mixed infection in tomato plants from the same crop field. This virus exhibited the typical genome organization of the DNA-A component of NW bipartite begomoviruses. Phylogenetic analysis showed that ToTLV is a member of the SLCV clade, although it was not closely related to other members within the group (Figure 3). Recombination evidence was detected in ToTLV by seven methods implemented in the RDP4 program. The recombination breakpoints were found around the intergenic region and the 5′ end of the *rep* gene. The N-terminal part of Rep and the adjacent intergenic region is a recombination hot-spot in the genome of begomoviruses [22]. ToYMoV and JacYVV were suggested as putative parents involved in the recombination event. JacYVV has been found infecting the weed *J. tamnifolia* in Zulia state, Venezuela [21] and ToTLV was uncovered in the same Venezuelan state. Relationships between begomoviruses infecting tomato and species of the family *Convolvulaceae* have been previously reported in Venezuela [23]. In this country, the round-year production and distribution of tomato seedlings throughout major tomato-producing states, as well as the wide predominance of polyphagous *B. tabaci* Middle East Asia Minor 1 [24,25], favors a dynamic pathosystem leading to diversification and evolution of begomoviruses.

Despite several attempts based on RFLP-RCA and PCR assays, no putative cognate DNA-B component could be found in association with the single component of ToLTV. Hence, the development of an infectious clone was required to determine the monopartite nature of ToTLV. Agroinoculation experiments on tomato and *N. benthamiana* plants confirmed that a single DNA component of ToTLV was required to establish the viral infection. The monopartite nature for an NW begomovirus was demonstrated in 2013 by two independent studies on ToLDeV [6,7]. To date, ToLDeV has been reported in Peru, Ecuador, and Brazil infecting only tomatoes. [2,5,6]. More recently, the infectivity of two monopartite begomoviruses from Brazil, *Tomato mottle leaf curl virus* (ToMoLCV) and Tomato leaf curl purple vein virus (ToLCPVV), was also demonstrated [26,27]. Tomato severe leaf curl virus (ToSLCV) has also been referred to as a monopartite NW begomovirus in Mexico and Central America [2]. Although ToSLCV infectivity has not been confirmed by inoculation of infectious clones so far, no evidence of the DNA-B component associated with ToSLCV has been found [28,29], even by using a high throughput sequencing strategy [30].

In addition to the typical five ORFs of the NW begomoviruses, ToTLV showed a sixth ORF partially overlapping the 5′ end of the CP gene. Interestingly, the AV2/V2 gene from OW begomoviruses is placed in the same region of the genome. This gene is associated with viral movement [31] and can act as an RNA silencing suppressor [32]. The ToTLV ORF 6 had a deduced amino acid sequence of 113 aa, while the size of AV2/V2 proteins range from 113 to 118 aa. However, no significant homology was found between the ORF 6 of ToTLV and any begomovirus protein in the GenBank database. Unlike ToTLV, an AV2-like gene was recently observed in a begomovirus, Sida golden yellow spot virus (SiGYSV), infecting *Sida* spp. in Brazil [33]. However, SiGYSV is considered to be an OW-like begomovirus as it shares a similar genome size with OW begomovirus (ca. 2800 bp), in addition to the presence of the AV2-like gene [33]. Ho et al. [34] estimated that deletions of more than 100 nt in the proximal promoter region of the AV2/V2 gene may lead to disruption of this gene in NW begomoviruses. In ToTLV, a putative transcript from ORF 6 was suggested by RT-PCR assays (Figure 5). Besides, using the criteria to identify a putative TATA box in the CP gene promoter region of geminiviruses [35], we have identified the sequence TATTAT at the position -39 from the potential star codon, which might represent a candidate for this transcription element. However, further studies will be needed to confirm whether this potential transcript is translated and its expression might affect the viral infection. In this work, we have found ToTLV infecting two tomato plants in Venezuela and new surveys in tomato fields will provide insight about the frequency and genetic diversity of this virus in the country.

Overall, our results along with previous studies raise new questions about viral movement in plants of NW monopartite begomoviruses in the absence of V2 protein and which mechanisms they use to develop a systemic infection.

## Figures and Tables

**Figure 1 viruses-11-00327-f001:**
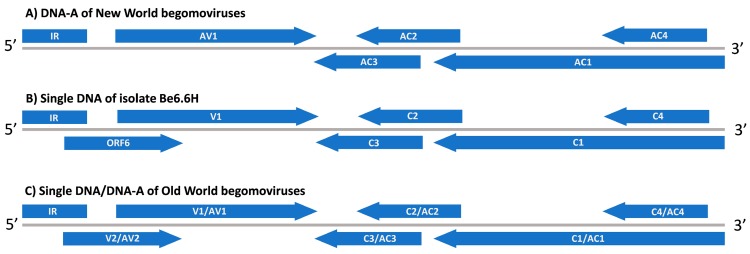
Genome organization of begomoviruses. (**A**) Typical genome organization of the DNA-A component from New World begomoviruses. (**B**) Genome organization of the clone Be6.6H displaying a sixth ORF encompassing the intergenic region (IR) and the V1 gene encoding the coat protein. (**C**) Typical genome organization of a single DNA or DNA-A component from monopartite and bipartite begomoviruses of the Old World. The gray lines represent the genome size of single DNA or DNA-A component of begomoviruses (ca. 2600–2800 bp). The arrowheads indicate the sense 5′–3′ to each gene.

**Figure 2 viruses-11-00327-f002:**
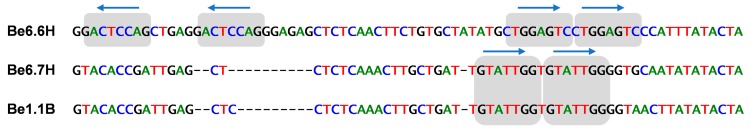
Iteron sequences found in the intergenic region of the clones Be6.6H, Be6.7H, and Be1.1B. Clone Be6.6H displayed different iterons as compared to clones Be6.7H and Be1.1B. Sequences in light grey boxes represent iteron sequences. Right arrows indicate forward repeats of iterons, and left arrows indicate inverted repeats of the iterons.

**Figure 3 viruses-11-00327-f003:**
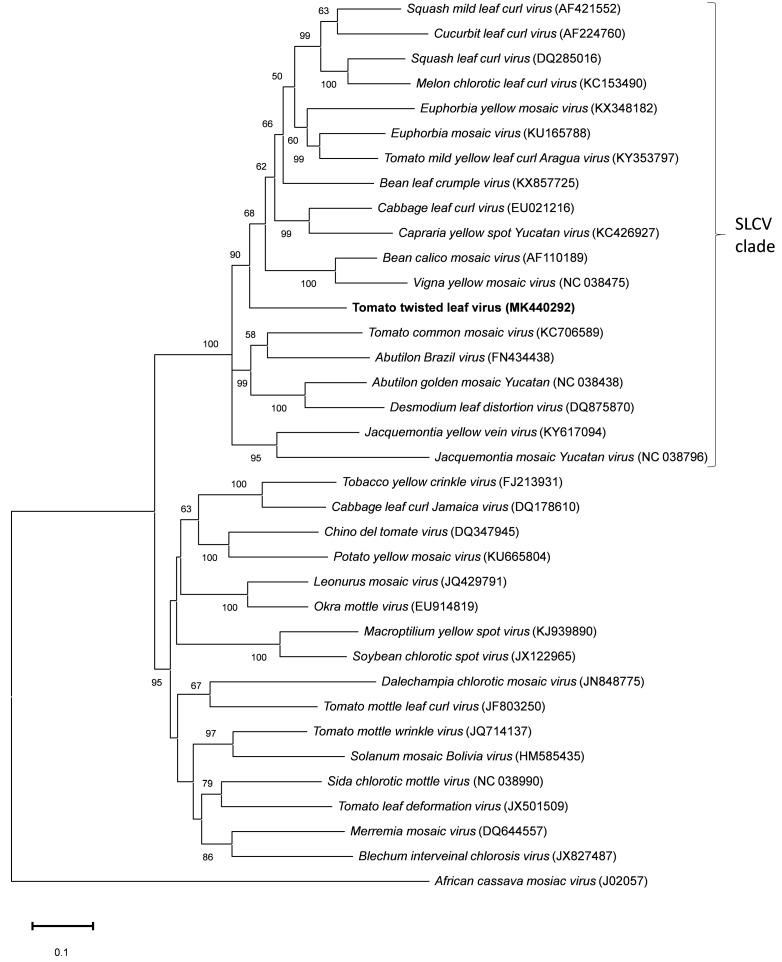
Phylogenetic relationships between the full-length DNA sequence of the novel begomovirus Tomato twisted leaf virus (highlighted in bold) and those of related begomoviruses. The phylogenetic tree is based on the Maximum likelihood method using GTR+G as a nucleotide substitution model. Bootstrap values (500 iterations) above 50% are indicated for each node. *African cassava mosaic virus* (ACMV) was used as an outgroup. GenBank accession numbers are indicated in brackets.

**Figure 4 viruses-11-00327-f004:**
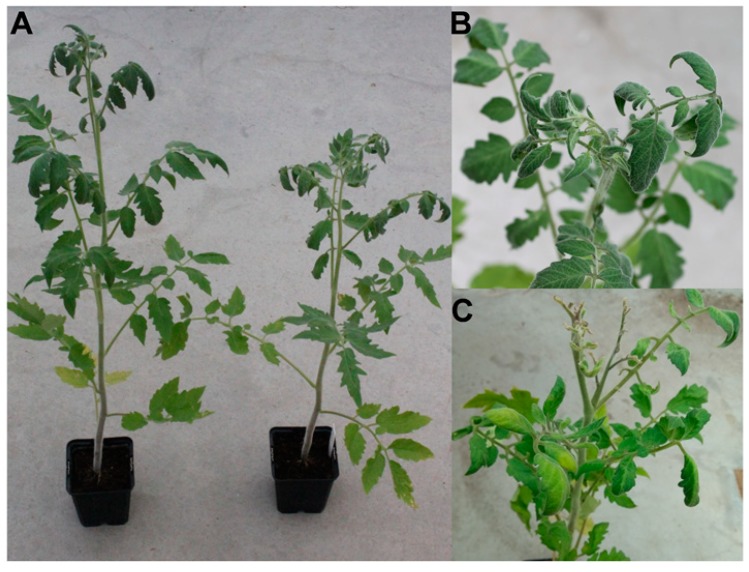
(**A**) Symptoms associated with ToTLV infection in tomato plants (cv. Moneymayker) after agroinoculation. Mock-inoculated (left) and ToTLV-inoculated (right) plants at four weeks post inoculation (wpi). (**B**) Twisted leaf symptoms observed in ToTLV-inoculated plants. (**C**) Severe symptoms (e.g., size reduction of young leaves, chlorosis, and leaf deformation) observed in tomato plants after eight wpi.

**Figure 5 viruses-11-00327-f005:**
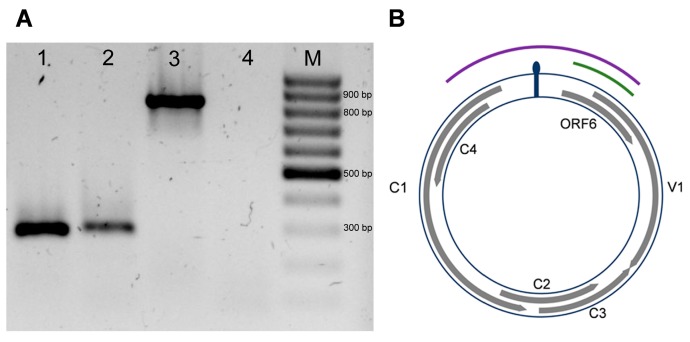
Genetic analysis of ORF 6 from ToTLV. (**A**) PCR and RT-PCR products from ToTLV-infected tomato plants. PCR and RT-PCR assays, respectively, using primers Be-089F/Be-363R (1 and 2). PCR and RT-PCR assays, respectively, using primers Be-2147F/Be-363R (3 and 4). (**B**) Schematic representation of ToTLV showing the typical five genes of the DNA-A components of New World begomoviruses (capsid protein gene (V1), replication-associated protein gene (C1), transcriptional activator protein gene (C2), replication enhancer protein gene (C3), and the C4 protein gene (C4)) and the additional ORF 6, which encompasses the intergenic region (IR) and the 5′ end of the V1 gene. The green line represents the expected amplification product with primer pair Be-089F/Be-363R. The purple line represents the expected amplification product with primer pair Be-2147F/Be-363R.

**Table 1 viruses-11-00327-t001:** Percent nucleotide sequence identity between Tomato twisted leaf virus (ToLTV) genome and closely related begomoviruses.

^1^ Begomovirus Species	ToTLV	EuMV	CabLCV	ToMYLCV	BLCrV	JacYVV
ToTLV (Venezuela)	100.0					
EuMV (Cuba)	82.4	100.0				
CabLCV (Mexico)	82.1	82.5	100.0			
ToMYLCV (Venezuela)	81.7	88.1	82.9	100.0		
BLCrV (Colombia)	81.2	82.6	83.3	84.1	100.0	
JacYVV (Venezuela)	81.1	81.0	79.3	79.8	78.9	100.0
EuYMV (Brazil)	80.7	82.3	80.7	84.6	80.5	77.3

^1^ Acronyms refer to Tomato twisted leaf virus (ToTLV; MK440292), *Euphorbia mosaic virus* (EuMV; KU165788), *Cabbage leaf curl virus* (CabLCV; EU021216), *Tomato mild yellow leaf curl Aragua virus* (ToMYLCV; JN368145), *Bean leaf crumple virus* (BLCrV; KX857725), *Jacquemontia yellow vein virus* (JacYVV; KY617094), and *Euphorbia yellow mosaic virus* (EuYMV; KY559488). Country of origin to each viral isolate is in brackets.

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
