# Peer review of "Tomato Twisted Leaf Virus: A Novel Indigenous New World Monopartite Begomovirus Infecting Tomato in Venezuela"

_viruses, 2019, doi:10.3390/v11040327_

Reviewer 1 Report

Virus discovery is an easy business now. A variety of methods have been improved to study plant virus biodiversity, including enrichment for virus-like particles or virus-specific RNA or DNA, or the extraction of total nucleic acids, followed by next-generation deep sequencing and bioinformatic analyses. Thus, many new viruses have been discovered, e.g. there are now 9 genera within the family Geminiviridae; a couple of years ago, there have been only four. Romay and colleagues found and investigated a new monopartite begomovirus in tomato. They fulfilled the Koch’s postulates and it’s proposed to name it tomato twisted leaf virus (ToTLV). Most interestingly, it has an additional ORF, which is located in the “V2 region”, but has no aa sequence similarity to any other protein in the public databases.

I do have some minor and major comments for this study:

-          L. 71, the samples showed leaf deformation and stunting symptoms – but did they show twisted leaves?

-          The PCR needs to be described in more detail. The authors used pGEM-Teasy as cloning vector, thus TA cloning with requires a A-overhang. Presumably, they used a polymerase without proof-reading, the Taq-Polymerase.

-          Are the acc.# (l.93) correct?

-          The RCA was only performed on sample# 332, but 3 isolates have been cloned, with different sequence, different iterons (Fig. 2). So, a mixed infection?

-          L.115 Has the RCA product only been digested with BamHI, EcoRI, HindIII and PstI? What if the DNA B circle has no such restriction sites. Jeske, 2018 (Viruses) describes there pretty nicely how it should be done – and then I’m convinced that there is no DNA B component.

-          L. 138, As negative control

-          Ten tomatoes and ten N.b. were inoculated with the infectious clone by agrobacteria, 3 biological replicates, but the authors don’t show any statistics – infection rate. No pictures of the N.b. infected plants. Figure 4 shows only tomato, but no details. Stunting has not been measured, again no numbers, no statistics. Looks like, the pictures have been made from above the plant. One should use a black background, make the picture from the side, in such a way, that one can see the growth difference. Detailed pictures, so that one can see symptoms clearly…

-          Figure 5: An RT-PCR is not sufficient. The authors need to provide 5’ and 3’ RACE data to show that ORF6 is transcribed. There will be two transcripts, one for ORF6, one for V1, if ORF6 is transcribed. Is there a TATA box upstream of the ORF6 and other elements one usually find upstream of an ORF? Is the ATG of the ORF6 in context with Kozak rules?

Good English, reads well, but not a comprehensive analysis to characterize a new virus, definitely not state-of-the-art (e.g. no deep sequencing).

Author Response

Comments and Suggestions for Authors

Virus discovery is an easy business now. A variety of methods have been improved to study plant virus biodiversity, including enrichment for virus-like particles or virus-specific RNA or DNA, or the extraction of total nucleic acids, followed by next-generation deep sequencing and bioinformatic analyses. Thus, many new viruses have been discovered, e.g. there are now 9 genera within the family Geminiviridae; a couple of years ago, there have been only four. Romay and colleagues found and investigated a new monopartite begomovirus in tomato. They fulfilled the Koch’s postulates and it’s proposed to name it tomato twisted leaf virus (ToTLV). Most interestingly, it has an additional ORF, which is located in the “V2 region”, but has no aa sequence similarity to any other protein in the public databases.

I do have some minor and major comments for this study:

-          L. 71, the samples showed leaf deformation and stunting symptoms – but did they show twisted leaves?

Response: The terms “leaf deformation” or “leaf distortion” have been already used to assign the scientific name of several tomato begomovirus species recognized by the International Committee on Taxonomy of Viruses (ICTV). Tomato plants infected with ToTLV showed this kind of symptom. Therefore, we have proposed the term “twisted leaf” to assign the scientific name for this putative new begomovirus species.    

-          The PCR needs to be described in more detail. The authors used pGEM-Teasy as cloning vector, thus TA cloning with requires a A-overhang. Presumably, they used a polymerase without proof-reading, the Taq-Polymerase.

Response: this has been done.

-          Are the acc.# (l.93) correct?

Response: It was corrected in the text.

-          The RCA was only performed on sample# 332, but 3 isolates have been cloned, with different sequence, different iterons (Fig. 2). So, a mixed infection?

Response: Yes, there was a mixed infection. This point was added in the text.

-          L.115 Has the RCA product only been digested with BamHI, EcoRI, HindIII and PstI? What if the DNA B circle has no such restriction sites. Jeske, 2018 (Viruses) describes there pretty nicely how it should be done – and then I’m convinced that there is no DNA B component.

Response: We agree that RCA-RFLP is not enough to demonstrate the absence of another DNA-B. Hence, we have used the same protocol followed by other independent laboratories (Melgarejo et al., 2013; Sanchez-Campos et al., 2013; Macedo et al., 2018) to show the monopartite nature of a begomovirus from the New World. These studies were based on RCA-RFLP, PCR-based detection of DNA-B using universal primers and development of infectious clones of the putative monopartite begomovirus. However, we think that a deep sequencing approach will be quite useful not only to verify the absence of DNA-B associated with this virus, but also to conduct more compelling analysis in further screening of this virus in the field.   

-          L. 138, As negative control

Response: this has been done.

-          Ten tomatoes and ten N.b. were inoculated with the infectious clone by agrobacteria, 3 biological replicates, but the authors don’t show any statistics – infection rate. No pictures of the N.b. infected plants. Figure 4 shows only tomato, but no details. Stunting has not been measured, again no numbers, no statistics. Looks like, the pictures have been made from above the plant. One should use a black background, make the picture from the side, in such a way, that one can see the growth difference. Detailed pictures, so that one can see symptoms clearly…

Response: The transmission rates of the virus in tomato and N. benthamiana were added in the text. We have also included a supplementary figure showing symptoms observed in N. benthamiana.

-          Figure 5: An RT-PCR is not sufficient. The authors need to provide 5’ and 3’ RACE data to show that ORF6 is transcribed. There will be two transcripts, one for ORF6, one for V1, if ORF6 is transcribed. Is there a TATA box upstream of the ORF6 and other elements one usually find upstream of an ORF? Is the ATG of the ORF6 in context with Kozak rules?

Good English, reads well, but not a comprehensive analysis to characterize a new virus, definitely not state-of-the-art (e.g. no deep sequencing).

Response: We agree that an RT-PCR is not enough to demonstrate the production of a transcript from the ORF 6. In the revised version of manuscript, we have used the term putative transcript from ORF 6 instead of transcript from ORF 6. We have identified a candidate TATA box at the position -39 upstream of the ORF 6. However, we have indicated in the text that other tests such as RACE assays and mutational analysis are required to confirm the production of the transcript predicted by bioinformatic analysis and RT-PCR assays.

Reviewer 2 Report

This is a well-written manuscript that describes a putative new monopartite begomovirus in tomato plants. The methodology used is sufficient and correct, and results support the conclusions. The starting material is 1 unique full-length viral clone obtained from dried leaf samples of two tomato plants collected in 2005 in Venezuela. What is known about the epidemiological relevance of this isolate, in the past and today? Has it subsequently been detected by the authors or others, in other tomato crops from Venezuela or elsewhere?

Author Response

Comments and Suggestions for Authors

This is a well-written manuscript that describes a putative new monopartite begomovirus in tomato plants. The methodology used is sufficient and correct, and results support the conclusions. The starting material is 1 unique full-length viral clone obtained from dried leaf samples of two tomato plants collected in 2005 in Venezuela. What is known about the epidemiological relevance of this isolate, in the past and today? Has it subsequently been detected by the authors or others, in other tomato crops from Venezuela or elsewhere?

Response: The samples analysed in this study were collected during a survey conducted in crop fields of Venezuela in the mid-2000s, aiming to detect begomoviral diseases by PCR. However, few samples have been completely sequenced, so far. Hence, little is known about the frequency and genetic diversity of ToTLV. In the manuscript we have highlighted the need of new genetic studies, including high-throughput sequencing, to better understand the epidemiological relevance not only of this begomovirus but also other begomoviruses which have been detected in the country.

Reviewer 3 Report

Authors described the characterization of new monopartite begomovirus fond in tomato plant in Venezuela with potentially an additional ORF. The paper is well constructed and I found only two editing error :

Line22 species

Line 285 established

Author Response

Comments and Suggestions for Authors

Authors described the characterization of new monopartite begomovirus fond in tomato plant in Venezuela with potentially an additional ORF. The paper is well constructed and I found only two editing error:

Line22 species

Response: this has been done.

Line 285 established

Response: this has been done.